# The Use of IoT for Determination of Time and Frequency Vibration Characteristics of Industrial Equipment for Condition-Based Maintenance

**Ihor Turkin** , **Viacheslav Leznovskyi** , **Andrii Zelenkov** * , **Agil Nabizade, Lina Volobuieva** and **Viktoriia Turkina**

Software Engineering Department, National Aerospace University "Kharkiv Aviation Institute",
61070 Kharkiv, Ukraine; i.turkin@khai.edu (I.T.); lieznovskiy@gmail.com (V.L.); agilnabizade@gmail.com (A.N.);
l.volobuieva@khai.edu (L.V.); v.turkina@khai.edu (V.T.)
* Correspondence: a.zelenkov@khai.edu; Tel.: +38-066-447-5807

**Abstract:** The subject of study in this article is a method for industrial equipment vibration diagnostics that uses discrete Fourier transform and Allan variance to increase precision and accuracy of industrial equipment vibration diagnostics processes. We propose IoT-oriented solutions based on smart sensors. The primary objectives include validating the practicality of employing platform-oriented technologies for vibro-diagnostics of industrial equipment, creating software and hardware solutions for the IoT platform, and assessing measurement accuracy and precision through the analysis of measurement results in both time and frequency domains. The IoT system architecture for industrial equipment vibration diagnostics consists of three levels. At the autonomous sensor level, vibration acceleration indicators are obtained and transmitted via a BLE digital wireless data transmission channel to the second level, the hub, which is based on a BeagleBone single-board microcomputer. The computing power of BeagleBone is sufficient to work with artificial intelligence algorithms. At the third level of the server platform, the tasks of diagnosing and predicting the state of the equipment are solved, for which the Dictionary Learning algorithm implemented in the Python programming language is used. The verification of the accuracy and precision of the vibration diagnostics system was carried out on the developed stand. A comparison of the expected and measured results in the frequency and time domains confirms the correct operation of the entire system.

**Keywords:** internet of things; digital platform; vibration diagnostics; calibration; accelerometer; industrial equipment; Allan variance



## 1. Introduction

Technological advancements in information technologies are progressively transforming our lives [1]. These changes demand an accelerated pace of management decision making [2,3]. As stated by the authors in [4], in the current scenario, producing an innovative product (or providing a service) that meets user requirements typically involves the integration of resources and competencies from multiple companies. The main finding from article [5] highlights the necessity for research and development in several crucial areas, including digital equipment maintenance and end-to-end automation, in order to enhance industries' preparedness for future problems.

Digital technologies and the Internet of Things (IoT) offer data homogeneity, distribution, editability, and the ability to self-reflect and reprogram, as stated in [6]. These features enable the implementation of multiple inheritance in distributed software applications, where no single owner possesses the entire design hierarchy or dictates the platform's core. As a result, products become open to new uses after manufacturing, as they can be arbitrarily combined through standardized interfaces [7].

The concept of condition-based maintenance (CBM) of industrial equipment [8] allows the determination of maintenance requirements and offers numerous benefits. CBM improves equipment credibility and dependability, and reduces maintenance resource costs compared to a late-scheduled maintenance approach. Under the CBM approach, maintenance is carried out only when specific metrics indicate declining performance or faults. The main problem with CBM is the need to spend significant resources on implementing equipment condition-monitoring tools, which usually include such non-invasive methods as visual inspection, and measurement of power consumption, noise, temperature, and vibration. Paper [9] proposes an integrated framework, which takes a broad perspective on CBM implementation, and integrates technological, organizational, and user-related elements. This study contributes to the field of CBM with a comprehensive view of implementation challenges and solutions in real-world implementations, from the original equipment manufacturer's (OEM's) point of view. Of the solutions proposed in article [9] for current research, we chose to prioritize the following in our work:

- Use a state-of-the-art IoT platform for development;
- Define modular project-level software decisions;
- Outline methods for collaboration between hardware and software specialists.

The authors of paper [10] emphasize the difference between Condition-based Maintenance (CBM) and Predictive-based Maintenance (PM) as two effective and complementary maintenance methods: CBM monitors the current condition; PM uses the CBM results to generate a future prediction for a machine.

Digital IoT platforms and end-to-end automation allow us to overcome the substantial resource consumption of the CBM concept. Well-known publications [11,12] extensively explore the development of cost-effective hardware and software solutions for vibration diagnostics using microelectromechanical systems (MEMS). A platform-oriented approach creates new possibilities for equipment fault diagnosis and state forecasting.

Smart sensors play a crucial role in CBM systems. According to the IEEE 1451.0-2007 standard [13], sensors with functions beyond the minimum required for measurements are classified as intelligent. Along with the digital interface and self-testing capabilities, these sensors have redundant functionality that simplifies their integration into networked applications.

In various mechanical systems, vibration diagnostics are an essential method for assessing the condition of mechanical systems, holding significant importance across multiple fields of application. Vibration is a highly versatile parameter that considers almost all aspects of a unit's state, allowing operating modes to determine the technical condition of the equipment.

Vibration acceleration is the vibration value directly related to the force that caused the vibration. Vibration acceleration characterizes the power interaction dynamic of elements inside the unit, which causes this vibration. The use of vibration acceleration is theoretically ideal since the accelerometer specifically measures the acceleration, which does not need to be specially converted. The disadvantage is that there are no practical developments regarding norms and threshold levels, and no generally accepted physical or spectral interpretations of the features of the manifestation of vibration acceleration.

Usually, for the vibration diagnostic, it is necessary to measure the vibration velocity. Vibration velocity is the speed of movement of the controlled point of the equipment during its precession along the measurement axis. Standard ISO 20816-1:2016 establishes general conditions and procedures for measuring and evaluating vibration, using measurements of vibration velocity root mean square value (RMS). The physical essence of the vibration velocity RMS parameter is the equality of the energy impact on the machine supports of an actual vibration signal and a fictitious constant, numerically equal in value to the RMS.

The accelerometer manufacturer establishes the output characteristics following extensive testing, typically encompassing the influence of various operating conditions, such as temperature changes and magnetic fields. Paper [14] proposes a method for estimating the thermal behavior of capacitive MEMS accelerometers and compensating for their drifts in

order to reduce orientation and temperature effects. It is a necessary solution, but insufficient for solving the general problem of compensating for accelerometer errors during regular operation.

The accelerometer metric of displacement at 0 g holds significant importance as it sets the baseline for measuring actual acceleration. Mounting the system with an accelerometer introduces additional measurement errors, which can arise due to stresses in the printed circuit board and the application of various compounds during mounting. As recommended in [15], we will calibrate after system assembly to exclude these errors.

The ISO 16063 series standards [16] set the modern requirements for vibration sensors and their calibration methods. Usually, a MEMS accelerometer calibration involves averaging the measurement values using a calibration scheme, where the accelerometer system is positioned to have one axis, typically the $Z$ axis, experiencing a 1 g gravitational field, while the other axes, $X$ and $Y$, remain in a 0 $g$ field. After installation at a specific location, additional calibration is conducted by comparing the measurement results with those of a reference accelerometer [17,18].

Although recommendations to use simultaneous analysis of vibration parameters in the time and frequency domains have been known since the standard [19], modern publications [20,21] do not attempt to combine analysis methods and the capabilities of modern IoT technologies into one whole.

Known hardware and software solutions for vibration diagnostics have the following disadvantages:

- Use of outdated sensors for measuring acceleration [22,23];
- Use of energy-inefficient wireless communication protocols between the sensor and the host [23];
- Use of suboptimal data processing algorithms that meet the standard [19] but are insufficient to ensure energy efficiency, affecting the duration of autonomous operation [12].

The study's objective is to offer a new resource-efficient IoT-oriented wireless solution and technology for vibration diagnostics, which utilizes the contact method of vibration measurement using MEMS accelerometers according to the standard [19].

To achieve the goal, we carry out the following tasks:

- Justify the choice of a cloud solution;
- Develop the architecture of a digital platform for vibration diagnostics, including a relational database model for storing measurement results;
- Determine the procedure for calibrating linear acceleration sensors;
- Propose a method for processing measurement results based on analysis in the frequency and time domains, which allows the application of a standard for assessing the current state of industrial equipment;
- As a result, prove that combining state-of-the-art information technologies and data analysis methods in the time and frequency domains allows us to achieve solutions qualitatively better than those known before.

## 2. Materials and Methods

### 2.1. IoT Platform Software and Hardware Solutions

Diagnostic results are processed and formed in a system with a three-level architecture (Figure 1). The main criterion in the development was to reduce installation and operation costs while ensuring high energy efficiency for extended autonomous operations. The system architecture consists of three levels:

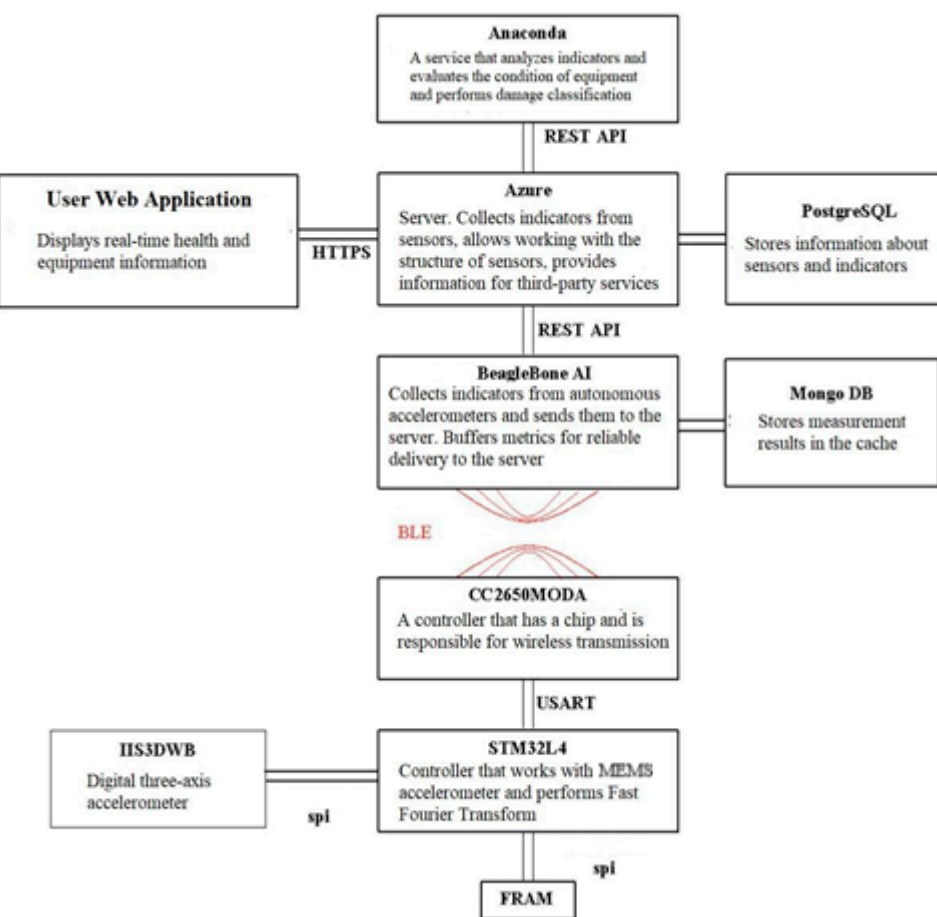

**Figure 1.** The architecture of the IoT system for vibration diagnostics.

1.  The autonomous sensor level is responsible for reading vibration acceleration indicators. It is based on an STM32L476 microcontroller and a three-axis digital accelerometer from STMicroelectronics called IIS3DWB [24]. The IIS3DWB capacitive accelerometer is mounted on the monitoring object and connected to the microcontroller via the Serial Peripheral Interface (SPI). It has low power consumption, high resolution (16 bits), and a reprogrammable measurement range of $\pm 2$ g, $\pm 4$ g, $\pm 8$ g, and $\pm 16$ g. The measurement result is read byte by byte in the form of 16-bit data. IIS3DWB has a bandwidth ranging from 0.05 to 6000 Hz, enabling it to capture vibrations with frequencies up to 1000 Hz. The IIS3DWB accelerometer saves the resulting calibration values in its OFFSET registers for automatic error compensation. Each OFFSET register's content is added to the measured acceleration value along the respective axis, and the resulting values are then stored in the DATA registers. Depending on the measurement frequency, these sensors are designed for autonomous operation for 6–12 months. We use BLE (Bluetooth Low Energy) digital wireless data transmission technology to transmit data from the sensor to the hub level, ensuring low energy consumption.

2.  The hub layer is deployed on a device using a single-board microcomputer called the BeagleBone® AI, which is designed to run artificial intelligence algorithms. This layer receives data from the autonomous sensor layer and transmits it to the server layer. Depending on the selected analysis algorithm, it may pre-process data before transmission, significantly reducing the server's load.

3.  The server layer offers an API that allows clients and third-party services to interact with the sensor structure and data. The Microsoft cloud platform Azure IoT Suite [25] provides infrastructure for creating and managing applications in the cloud. The Azure Internet of Things Suite is a comprehensive service that leverages the full

capabilities of Azure to establish connections with devices. It effectively captures a diverse range of data generated by these devices. Through seamless integration and organization, the suite manages and analyzes the data, presenting it in a format that facilitates informed decision-making. We selected Microsoft Azure IoT for further implementation because this platform provides well-established solutions, and its budget requirements are feasible for startups in the initial phase.

### 2.2. Digital Platform Database

The PostgreSQL DBMS was selected to ensure long-term data storage in the digital platform for vibration diagnostics of industrial equipment at the second level of the hub. PostgreSQL has been in development since 1996. PostgreSQL boasts advanced features, including Multi-Version Concurrency Control, asynchronous replication, and nested transactions (savepoints). At the third level of the server platform, we use the NoSQL database MongoDB for caching data on the BeagleBone AI Mini PC.

The ER model of the digital platform database (Figure 2) characterizes the relationships between the following entities: sensors, IoT devices in which they are included, and data on the results of measurements by these sensors.

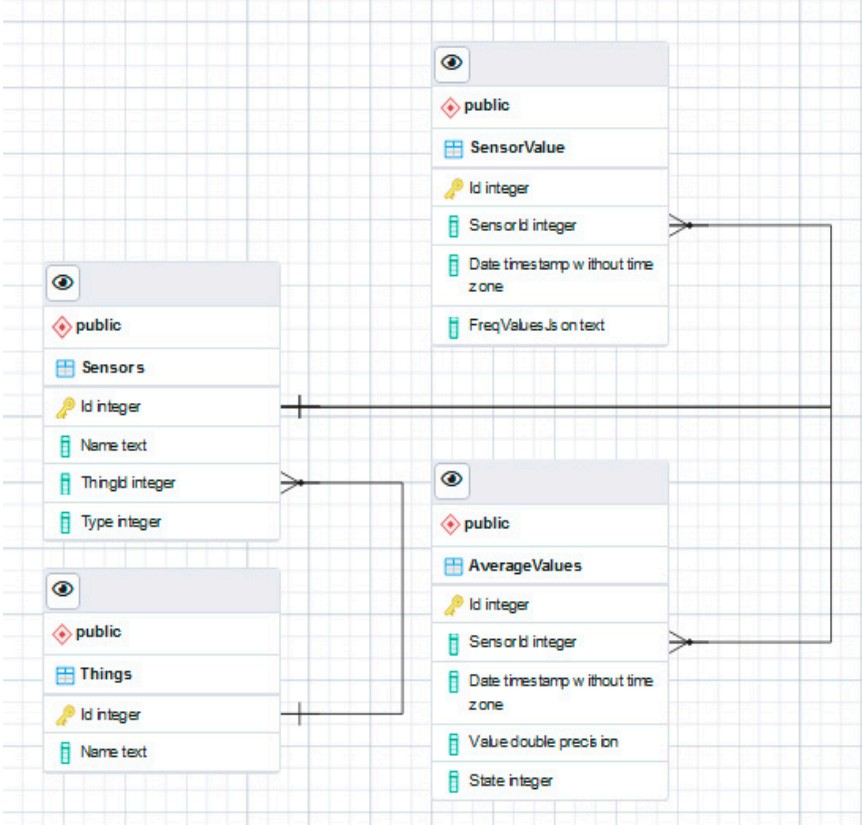

**Figure 2.** ER model of the database of the digital platform.

Entities in the database, which are the basis for business logic, contain the following data according to the IEEE 1451.0-2007 standard:

- AverageValue—average values of measurement results by sensors, such as average rotation speed;
- Sensor—data about sensors: sensor identifier, name;
- SensorValue—data on current and historical measurement results: timestamp (Timestamp) and result (Value);
- Things—data about the equipment on which the sensors are installed: name and location.

- The definitions of relationships between entities are presented as follows:
- The «Sensor» entity is related to the «AverageValue» entity by the ratio «1: N»—each sensor has many numerical indicators;
- The «Sensor» entity is related to the «SensorValue» entity by a «1: N» relationship—each sensor has many measurement results;
- The «Thing» entity is related to the «Sensor» entity by a «1: N» relationship—each piece of industrial equipment can have many sensors.

### 2.3. The Bench Equipment and Measurement Algorithm

The adequacy of the system, the accuracy, and the correctness of the work of the components were evaluated using the bench equipment shown in Figures 3 and 4. This setup replicated an electric motor and generator interconnected by a coupling.

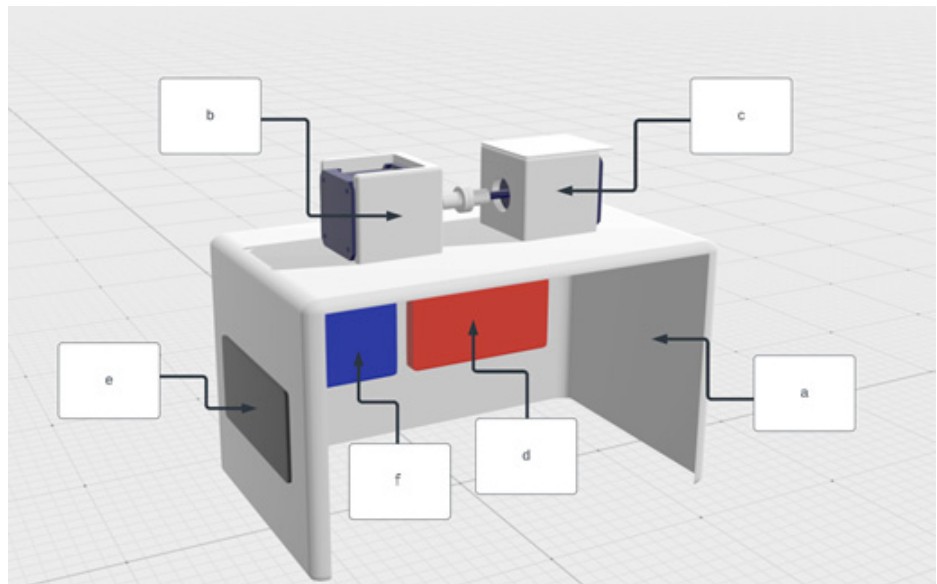

**Figure 3.** A 3D-model of the testbed (a—Testbed base, b—Load node, c—Node with motor and vibration sensor mount, d—Power supply unit, e—User control panel, f—Testbed controller unit).

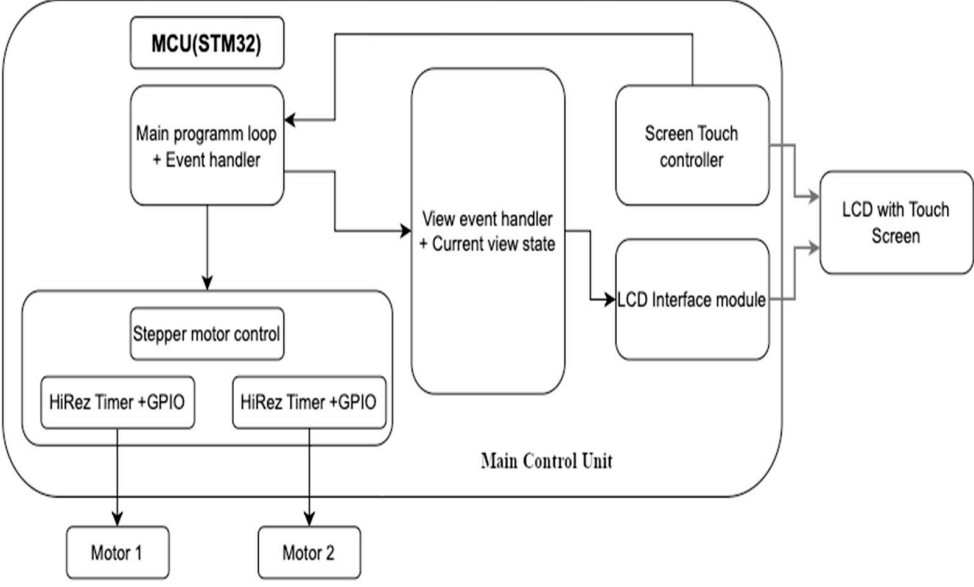

**Figure 4.** Schema of the testbed electronics.

The testbed replicates the operational conditions of the pumping unit at the enterprise. The testbed consists of:

(a)　A testbed base;
(b)　A load node that replicates the behavior of the generator;
(c)　A test node that replicates the behavior of the electric motor, with a vibration sensor mount;
(d)　A power supply unit which provides power for the whole testbed;
(e)　A user control panel with a sensor display for visualization and control of motor speed;
(f)　A testbed controller unit, which controls motors and the user control panel.

The central controller is the STM32F429 chip; the primary actuators are stepper motors in conjunction with stepper motor controllers. High-precision timers are used as part of the microcontroller to generate pulse sequences. The use of hardware timers allows for minimization of the load on the central core of the controller since, after the initial initialization, it will be sufficient for the controller to change the value of the divider register when a speed change event occurs.

The operation of the microcontroller is based on a mechanism for responding to events coming from the stand operator when interacting with the touch screen. When registering a click on an interface element, the main loop will update the current state of the GUI as well as the value of the speed variables of the motors.

Figure 5 shows a measurement system algorithm.

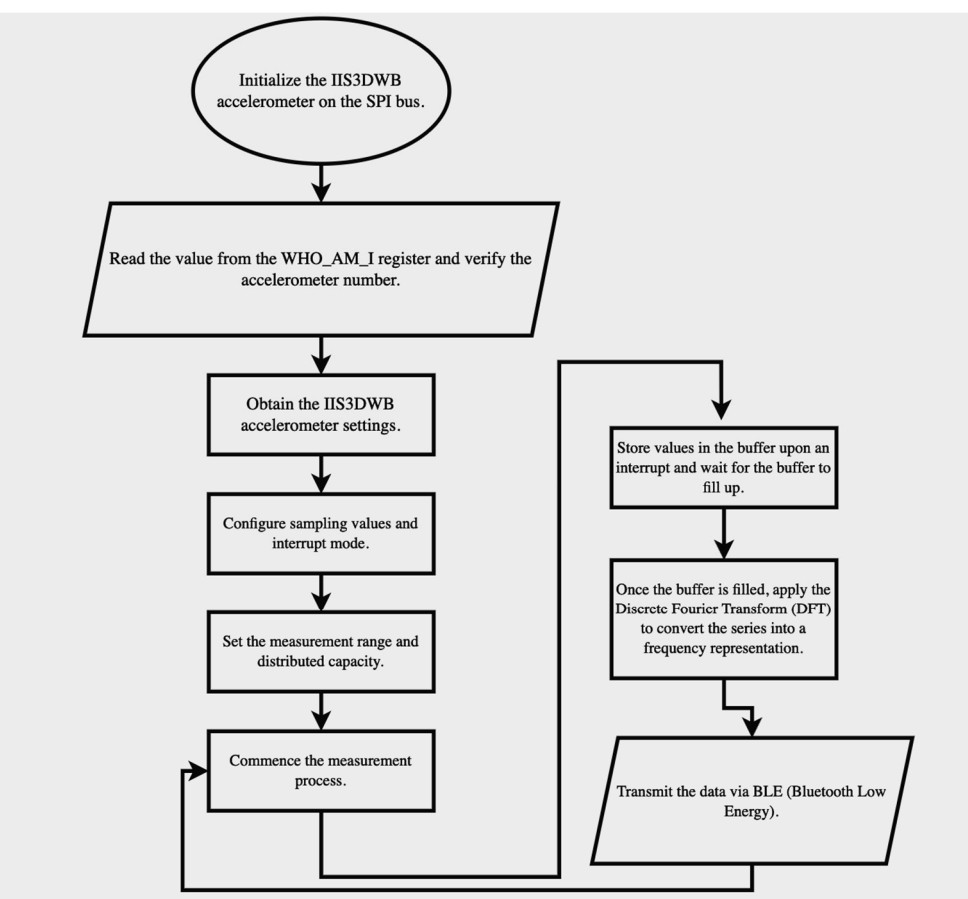

**Figure 5.** Measurement algorithm.

According to the algorithm, the most energy-consuming steps will be steps of the measurement process and the transmission of data via BLE.

### 2.4. Sensor Calibration Procedure

Sensor calibration is divided into two steps:

- Measuring energy consumption in all potential work conditions;
- Lengthy testing of sensors in field conditions.

However, we must first check the calibration with debugging using the JTAG interface. The sensor has three axes, so the calibration is carried out under conditions in which one axis is installed perpendicular to the plane of the work table.

If the *Z* axis is perpendicular to the plane of the desktop (Figure 6), then the measurement results at rest along the *Z* axis will equal 1 *g*, and 0 *g* along the *X* and *Y* axes. Two identical designs along the measurement axes can be made perpendicularly to obtain a biaxial accelerometer, but the third axis, usually vertical-*Z*, requires a different construction. This results in poorer performance for the third axis, reducing sensitivity and increasing error and noise [24].

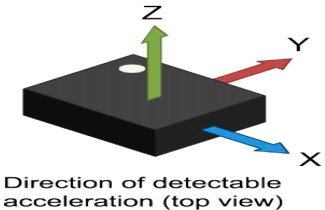

**Figure 6.** Scheme of the location of measurement axes.

Figure 7 presents the measurement results of the sensor calibration.

| 📊 Build Analyzer | ≡ Static Stack Analyzer | 🐞 Debug | 🖩 SFRs | 🔲 Memory | ☰ Include Browser | 👓 Expressions ⊠ |
|---|---|---|---|---|---|---|
| **Name** | | | **Value** | | | |
| ᵡ⁺ʸ "id" | | | | | | |
| ᵡ⁺ʸ "x" | | | 0.021 | | | |
| ᵡ⁺ʸ "y" | | | 0.014 | | | |
| ᵡ⁺ʸ "z" | | | 1.015 | | | |
| ➕ Add new expression | | | | | | |

**Figure 7.** Measurement results during calibration.

### 2.5. The Method of Processing Measurement Results

The digital platform offers customers an equipment condition assessment in the form of classification, categorized as "good", "satisfactory", or "unsatisfactory". The primary goal of developing the evaluation algorithm is to fulfill the minimum training requirement by using the smallest amount of labeled data. As a result, the algorithm should be capable of recognizing patterns and learning from unlabeled input data, a technique known as System Identification. The algorithm was implemented using the Python programming language. Python allows for the use of various tools and libraries for machine learning, data processing, and signal analysis. The equipment condition evaluation algorithm was split to separate services to provide greater scalability and flexibility for the whole system.

Accelerometer noise analyses using the Allan variance method include the following steps:

1. Primary statistical data processing.

For statistical processing of the initial information, the correspondence of the data to the normal distribution or Gaussian distribution of the probability density distribution is checked at the first stage.

For all points, the arithmetic mean measurement at each position is calculated. We start by using the average. The scatter is indicated by the standard deviation (SD). The mean deviation is the spread around the mean and shows the initial instability.

$$x = \frac{1}{n}\sum_{j=1}^{n} x_j, \, SD = \sqrt{\frac{1}{n}\sum_{j=1}^{n}\left(x_j - x\right)^2}, \tag{1}$$

where *n*—the total number of measurements;

$x_j$—measurement results of j points.

   For groups of positions in the runs, the overall average across all runs and the average variability between runs are calculated, considering the exclusion of anomalous measurements. To assess compliance with the Gaussian distribution, we utilize the Shapiro–Wilk test [26]. This test employs the sum of squared deviations between the characteristic functions derived from sample data and a normal distribution.

2   Allan deviation analysis provides excellent differentiation between the particular noise types. We use the Allan transform to quickly check whether the measurement results are consistent with «white» noise, as proposed in the paper [26]. The formula for calculating the Allan variance $\sigma_A^2(\tau)$ under the condition of a uniform polling step $\Delta t$ is as follows:

$$\sigma_A^2(\tau) = \frac{\sum_{n=0}^{L-2l} \left( \sum_{i=1}^{l} ((x(t_{n+i}) - x(t_{n+l+i}))) \right)^2}{2\tau^2(L - 2l + 1)} \tag{2}$$

where $L$—the total number of measurements;

$l$—total number of measurements in the averaging interval $1 < l \leq L/2$;
$x(t_k)$—measurement result at time $t_k = k \bullet \Delta t$.

3   For noise spectral density analysis:

$$f(x) = a_0 + \sum_{n=1}^{\infty} \left( a_n \cos \cos \frac{n\pi x}{L} + b_n \sin \sin \frac{n\pi x}{L} \right), \tag{3}$$

where $a_n$, $b_n$—Fourier series coefficients.

We use Discrete Fourier Transform (DFT), which realizes transformations:

$$X_k = \sum_{j=0}^{n-1} x_j \left( \cos\left( \frac{2\pi}{n} kj \right) - i \bullet \sin\left( \frac{2\pi}{n} kj \right) \right) \tag{4}$$

   We realized the Discrete Fourier Transform (DFT) at the second (hub) level to enhance determination of the frequency composition of input signals. The external software library (FFTW) was used to compute the frequency vibration characteristics. The DFT recorded spectral information represents the vibration acceleration signals of each axis. To improve the accuracy of the Fourier transform, we used the adaptive filtering method for the MEMS gyroscope with a dynamic noise model [15].

4   One of the main reasons for the accelerometer's systematic errors is the temperature-dependence of its characteristics [24]. When integrating to calculate the vibration speed, these errors quickly accumulate and give an utterly unreliable result. To determine the root-mean-square of the vibration velocity ($RMS_{VV}$), as recommended by standard ISO 20816-1:2016, we use the integration of the measured vibration acceleration ($VA$) with a correction for the moving average of the vibration acceleration ($MA_{AV}$) and vibration velocity ($MA_{VV}$):

$$MA_{VA_k} = g \bullet \sum_{j=k-n}^{k+n} x_j/(2 \bullet n + 1), \ VA_k = g \bullet x_k - MA_{VA_k},$$
$$V_k = V_{k-1} + (VA_{k-1} + 4VA_k + VA_{k+1})\Delta\tau/6 \tag{5}$$
$$MA_{VV_k} = \sum_{j=k-n}^{k+n} V_j/(2 \bullet n + 1), \ VV_k = V_k - MA_{VA_k}$$

where $g$—gravitational acceleration, $g$ = 9.8 m/s$^2$;

$x_j, x_{k-1}, x_k, x_{k+1}$—raw acceleration data, $g$;
$VA_{k-1}, VA_k, VA_{k+1}$—calculated values of the vibration acceleration with acceleration offset compensation, m/s$^2$;

$V_k$, $V_j$—calculated values of the vibration velocity without velocity offset compensation, m/s$^2$;

$VV_k$, $VV_j$—calculated values of the vibration velocity with velocity offset compensation, m/s$^2$;

$n$—variable that specifies the half-width of the time window for which averaging and offset compensation is performed.

With the known dependence of the vibration velocity on time $VV(\tau)$, root-mean-square value over a time interval $RMS_{VV}$ can be calculated as:

$$RMS_{VV}(\tau) = \sqrt{\frac{1}{T}\int_{\tau-T/2}^{\tau+T/2} VV^2(t)dt} \approx \sqrt{\sum_{j=k-n}^{k+n} VV_j^2/(2n+1)}. \tag{6}$$

### 2.6. Comparison with Analogs

We compared our decision with those previously offered by viewing the following three similar systems, each allowing several accelerometers to connect with a high-level system.

1.  The adaptive Kalman filter and Allan variance method used for Inertial Measurement with Unit MPU6050. This method is based on the dynamic noise model [22].
2.  The compact and low-powered MEMS accelerometer and microcontroller with wireless connectivity and a run time of approximately eight hours [23].
3.  A budget-friendly vibration measurement system dedicated to assessing the condition of construction structures [12].

## 3. Results

The sensors are calibrated to assess the potential of the hardware part for maintenance-free use. During calibration, we compared the results from the visualization tools and the debugger with the information displayed by the software.

1.  Using Allan variation, it is easy to eliminate the systematic error in estimating the statistical characteristics of the original series, while for uncorrelated data, the variance estimate will be unbiased. The presence of any periodic function will be displayed on the plot of the Allan variance versus time (Figure 8).

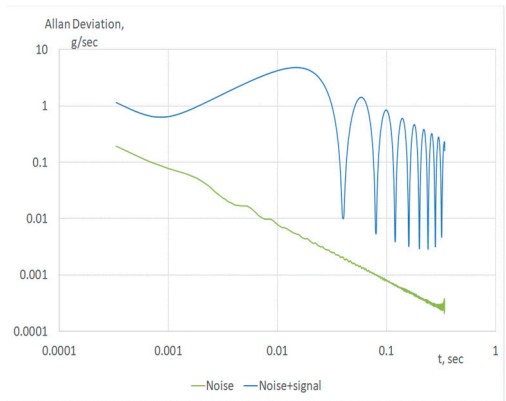

**Figure 8.** Simulation results: Allan deviation for white Gaussian noise and the additive mixture of 25 Hz harmonic signal and white Gaussian noise (signal-to-noise ratio is 40 dB).

Practically, the offset of the fitting straight line and the standard error of the deviation from the straight-line equation are reliable indicators of the magnitude of the vibration (Figure 9). As a result, we get visual evidence that the measurement noise is white noise as the slope of the characteristic on a log–log scale is −1. The problem with this solution is the lack of generally accepted standards linking the Allan deviation measurement results and the technical condition of industrial equipment.

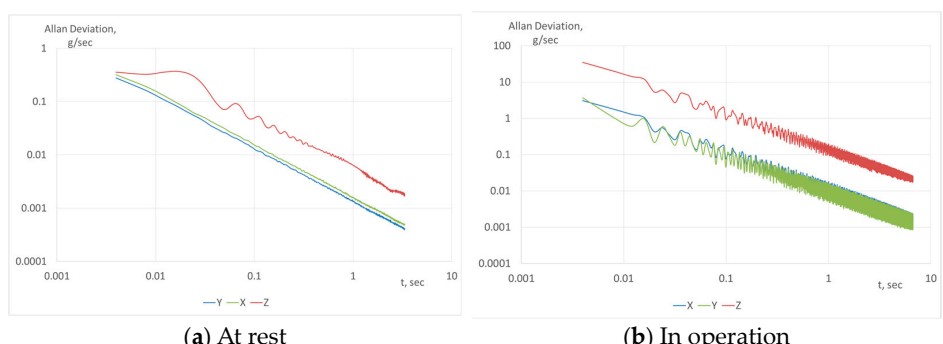

**Figure 9.** Allan deviation acceleration data at rest and during the operation of industrial equipment.

2. The noise spectral density analysis allows us to study the problem in the frequency domain. In the numerical simulation, the dependence of the results of the Fourier transform in the presence of an additive mixture of a single valid periodic signal (50 Hz) and noise of various intensities looks like Figure 10.

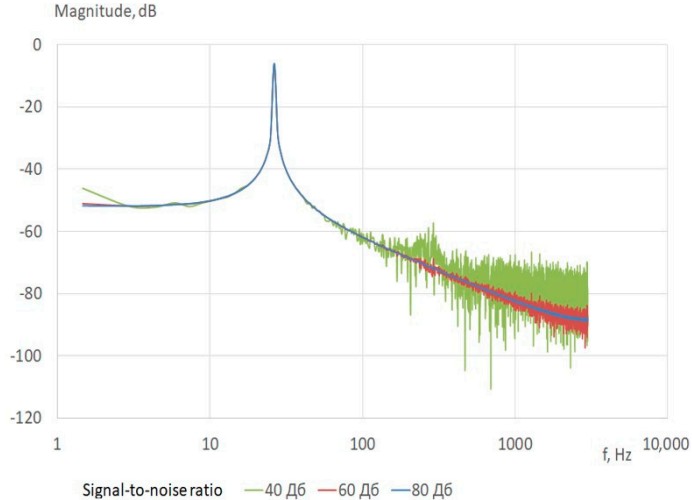

**Figure 10.** Outcomes of the simulation: The vertical acceleration frequency response under varying noise-to-signal ratios.

In contrast to theoretical model calculations, the results of practical measurements demonstrate the presence of many harmonics in the frequency range of 20–100 Hz (Figure 11). As a result, we concluded that the width of the 5–20 ms time window, determined by *n* in Equation (5), is the most suitable for calculating the moving average.

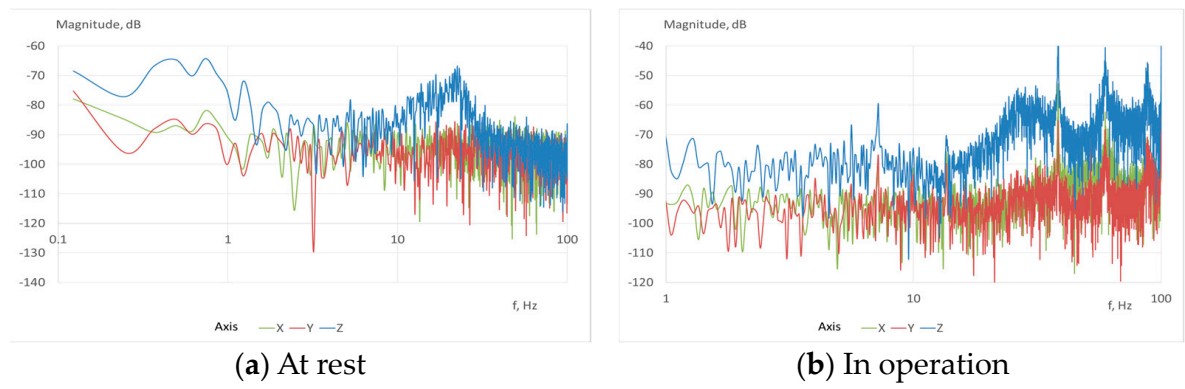

**Figure 11.** Frequency response of-3-axis accelerations with the different signal-to-noise ratios.

3. Despite pre-calibration, the raw data contains a significant bias error. With the standard placement of the MEMS accelerometer, the sensor along the *z*-axis, which is the least accurate due to the technological constraints of production, measures the value of the proper acceleration (Figure 12a). This error is integrated without compensation (5), leading to unreliable results. Moving average compensation removes this bias (Figure 12b).

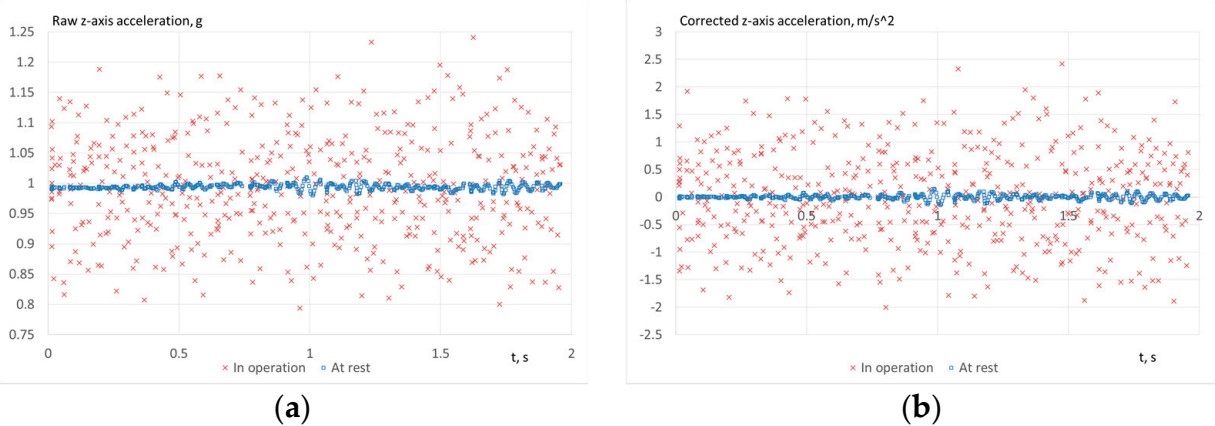

(**a**)  (**b**)

**Figure 12.** Z-axis acceleration data at rest and during the operation of industrial equipment: (**a**) row measurements: $x_k$ (5), (**b**) calculated values with acceleration bias compensation: $VA_k$ (5)).

Similar to compensating for the acceleration measurement error by using a moving average, compensating for the velocity calculation bias makes it possible to obtain more accurate vibration velocity estimates (Figure 13).

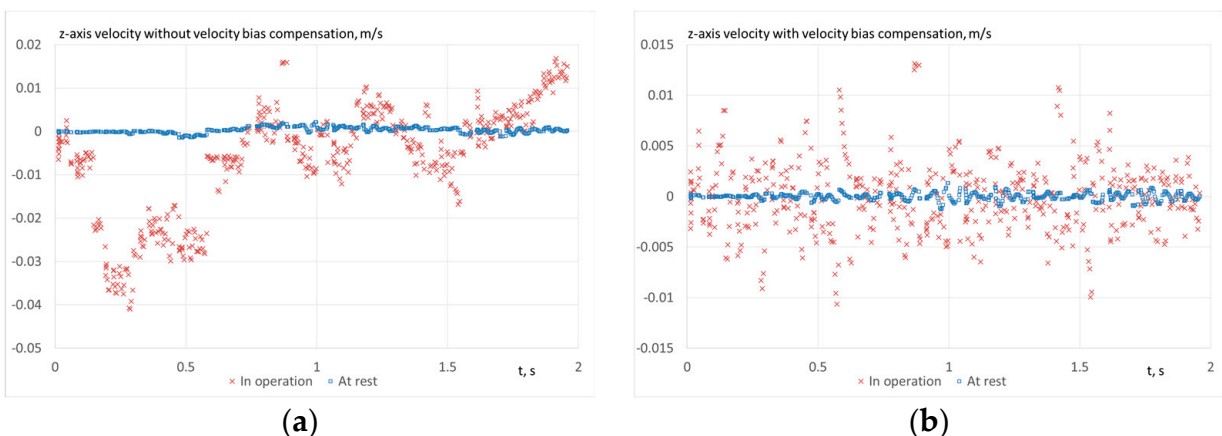

(**a**)  (**b**)

**Figure 13.** Velocity data at rest and during the operation of industrial equipment: (**a**) without velocity bias compensation, (**b**) with velocity bias compensation.

4. The vibration velocity RMS calculation according to Equation (6) gives the following results. The algorithm given in Section 2.5 confirms that the controlled industrial equipment is in good condition. The calculated RMS value of the vibration velocity corresponds to the recommendations for determining the boundaries of the vibration state zones according to the standard ISO 20816-1:2016. As a result, we have the following:

- The standard sets the limit of normal operation for low-power systems as "vibration velocity no more than 0.6 mm/s";
- With a time window width of $T = 0.1$ s, the measuring system fixes velocity at no more than 0.05 mm/s at rest and no more than 0.35 mm/s during operation (Figure 14);

- Extending the time window width *T* from 0.1 s to 1.6 s does not significantly improve the vibration velocity estimations (Table 1). It can be assumed that the time window length should be within 0.1–1 s. The experimental estimates of Allan's variation (Figure 9) and the results of the Fourier transform (Figure 11) fully correspond to this decision.

**Table 1.** Dependence of the calculated maximum vibration velocity (Max(RMS), m/s) on the time window width (*T*), during operation.

| *T*, s | Max(RMS), m/s |
|---|---|
| 0.1 | 0.35 |
| 0.2 | 0.28 |
| 0.4 | 0.26 |
| 0.8 | 0.25 |
| 1.6 | 0.25 |

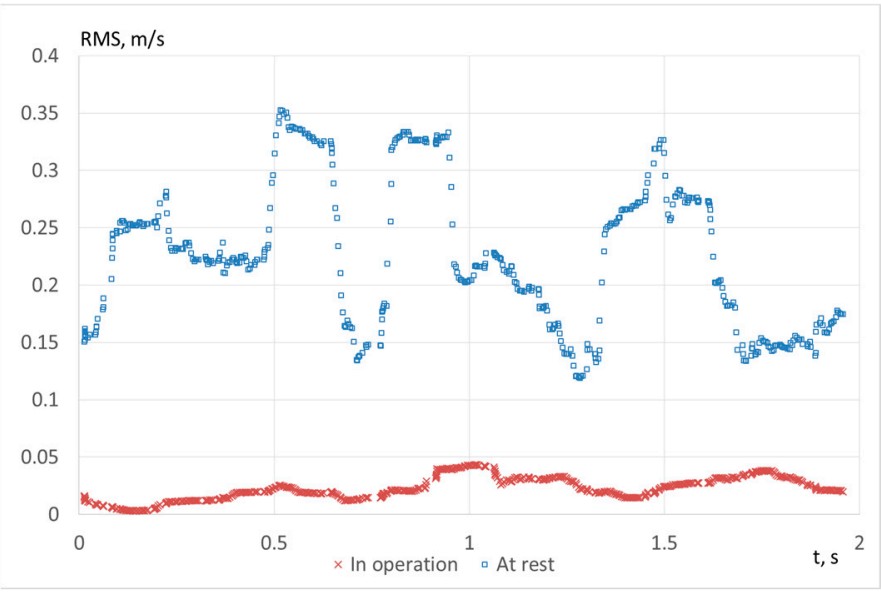

**Figure 14.** RMS of the vibration velocity with time window width *T* = 0.1 s.

## 4. Discussion

The industrial equipment condition-based and predictive-based maintenance concepts demand the complex use of multilevel digital technologies. The proposed software and hardware IoT solutions for vibration diagnostics, with the method of processing measurement results, appear to be a practical approach to accuracy and precision control of the technical state of industrial equipment.

Comparing our system with previously developed ones, which are referenced in the Introduction section, reveals the following advantages:

We achieve low-cost three-level hardware and software solutions for vibration diagnostics, with vertical integration of measuring devices with the Azure Internet of Things Suite digital platform;

- We utilize the specialized IIS3DWB smart accelerometer from STM, designed explicitly for vibration diagnostics, enabling an increased frequency range of measurements up to 6000 Hz.
- Using low energy consumption BLE and the STM32 microcontroller, we obtain a long period of autonomous work (up to one year). For comparison, prototype systems need to recharge every 8–12 h.

- We offer a method for integrating the measured vibration acceleration with a correction for the moving average of the vibration acceleration and vibration velocity, to move from the directly measured vibration acceleration to the vibration velocity RMS value recommended by the ISO 20816-1:2016 standard.

Consequently, we substantially reduce operating expenses by maintaining almost the same equipment and installation costs as in wireless analogs.

## 5. Conclusions

The proposed technique can effectively realize the contact method of vibration measurement using MEMS accelerometers according to the standard IEEE Recommended Practice for Inertial Sensor Test Equipment, Instrumentation, Data Acquisition, and Analysis.

Our choice of cloud solution, Microsoft Azure IoT, can be justified by the sufficient capabilities of the platform and Microsoft's pricing policy, which favors the possibility of startup projects. The developed architecture of the digital vibration diagnostics platform (Figure 1) and the relational database model (Figure 2) for storing measurement results provides an additional option for solving Predictive-based Maintenance tasks. The developed technological equipment for calibrating vibration acceleration sensors (Figures 3 and 4) allows us to reduce systematic and random accelerometer setting errors. The implemented method for the interpretation of measurement results in the frequency and time domains (Section 2.6) allows the application of the IEEE Recommended Practice for Inertial Sensor Test Equipment, Instrumentation, Data Acquisition, and Analysis requirements in assessing the current state of industrial equipment.

Thus, we proved that combining well-known and classical analysis methods in the time and frequency domains with the possibilities of their multilevel processing in the system allows us to achieve qualitatively improved results.

In the maturing of the project, we plan to maintain the limitations of the low cost of development and operation to solve the following tasks:

- Optimal distribution of solved tasks across the three levels of the IoT system;
- Reduction of the computational complexity of the algorithms in order to execute them at a lower level, increase autonomy at a low level and reduce the load on communication channels.

The upcoming stage of our plan encompasses the creation of supplementary microservices. These will facilitate the application of time series analysis techniques and state-of-the-art AI technologies, enhancing the quality of predictive maintenance. We plan to investigate, using mathematical models, the dependence of the leading indicators of Allan deviation of the vibration acceleration data on the technical condition of the equipment, measured through the root-mean-square of the vibration velocity in the theoretical part.

**Author Contributions:** Conceptualization, I.T. and V.L.; methodology, I.T., A.Z. and V.L.; software, V.L. and A.N.; validation, I.T. and V.T.; formal analysis, I.T. and L.V.; investigation, V.L.; data curation, A.Z. and L.V.; writing—original draft preparation, V.L.; writing—review and editing, A.Z.; visualization, V.T.; supervision, I.T.; project administration, I.T. All authors have read and agreed to the published version of the manuscript.

**Funding:** This research received no external funding.

**Data Availability Statement:** Data not available due to confidentiality.

**Conflicts of Interest:** The authors declare no conflict of interest.

### Nomenclature

| | |
|---|---|
| $x$ | Arithmetic means of the measurements |
| $\sigma_A^2(\tau)$ | Allan variance |
| $\Delta t$ | Time between measurements |
| $L$ | Total number of measurements |
| $l$ | Total number of measurements in the averaging interval $1 < l \leq L/2$ |
| $x(t_k)$ | Measurement result at the time $t_k = k \bullet \Delta t$ |
| $a_n, b_n$ | Fourier series coefficients |
| $g$ | Gravitational acceleration |
| $x_j, x_{k-1}, x_k, x_{k+1}$ | Raw acceleration data, $g$ |
| $VA_{k-1}, VA_k, VA_{k+1}$ | Calculated values of the vibration acceleration with acceleration offset compensation, m/s$^2$ |
| $V_k, V_j$ | Calculated values of the vibration velocity without velocity offset compensation, m/s$^2$ |
| $VV_k, VV_j$ | Calculated values of the vibration velocity with velocity offset compensation, m/s$^2$ |
| Abbreviations | |
| BLE | Bluetooth Low Energy |
| IoT | Internet of things |
| CBM | Condition-based maintenance |
| PM | Predictive Maintenance |
| MEMS | Micro-electromechanical systems |
| RMS | Root mean square |
| SPI | Serial Peripheral Interface |
| MA | Moving average |
| SD | Standard deviation |
| I$^2$C | Inter-Integrated Circuit |
| VA | Vibration acceleration |
| ADC | Analog-to-digital converter |
| DBMS | Database management system |
| DFT | Discrete Fourier transform |

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
