# Peer review of "The Use of IoT for Determination of Time and Frequency Vibration Characteristics of Industrial Equipment for Condition-Based Maintenance"

_computation, doi:10.3390/computation11090177_

Round 1

Author Response

Comments

Answers

- At the end of Section 1, the main contributions of the paper should be compared with the existent literature. It is essential that the authors demonstrate the quality or efficiency of their results, compared to well-established methods.

We took into account the comments and significantly changed the section, better structured it and somewhat shortened it, provided comments on the abbreviations, and at the end we formulated the aim of the article

- In Figure 3, please indicate the main components of the test stand. Furthermore, please provide some technical data of these components.

The photo of the stand has been replaced with its 3D model with the indication of its components. Some technical data of these components were added.

- In Figure 4, please indicate the main components. If the figure was not done by the authors, please provide the reference.

The Figure 4 is not fundamentally important in the general topic of the article, it has been deleted.

-The Conclusion section is sooner a summary of the work, without really presenting the conclusions that can be drawn from this research. Therefore, the quantitative results are required, and the meaningfulness of this study would be emphasized rather than presenting a summary on the technical works. Furthermore, the section should be extended with:

ü  the weak points of the proposed method;

ü  limitations of the method;

ü  Further studies.

The conclusion section was entirely rewritten.

Reviewer 2 Report

The paper is well written and the English language is good. The author takes a long time just describing the instruments and equipment that he used in the paper with its serial number. This will make the paper so long and distract the reader attention. The paper needs major revision and my comments are in below:

(1)    The introduction section is nor written in the right way since there is no aim of this paper at the end of this introduction.  The introduction section has many abbreviations which need more explanation.

(2)    The paper needs nomenclature especially for equations 1-5.

(3)    The author used Fast Fourier Transform of power spectrum analysis. Did the author take velocity or acceleration in this analysis?

(4)    The results show that the author use the Gaussian noise of signal to noise ratio. The signal to noise ratio has positive and negative signals please explain.

(5)    In the discussion section of the results, the author mentioned that he used the experiment results just to test the theoretical results. Could you please explain this in the result section?

The conclusion section looks like the abstract section and there is no real conclusion. The conclusion section should not contain reference number. Moreover, the discussion section should not contain reference number.

Author Response

Comments

Answers

(1)    The introduction section is nor written in the right way since there is no aim of this paper at the end of this introduction.  The introduction section has many abbreviations which need more explanation.

We took into account the comments and significantly changed the section, better structured it and somewhat shortened it, provided comments on the abbreviations, and at the end we formulated the aim of the article

(2)    The paper needs nomenclature especially for equations 1-5.

The nomenclature for equations 1-5 is significantly supplemented

(3)    The author used Fast Fourier Transform of power spectrum analysis. Did the author take velocity or acceleration in this analysis?

The Fast Fourier transform was performed for raw acceleration data.

(4)    The results show that the author use the Gaussian noise of signal to noise ratio. The signal to noise ratio has positive and negative signals please explain.

Signal-to-noise ratio is defined as the ratio of signal power to noise power. A ratio higher than 1:1 (0 dB) indicates more signal than noise.

(5)    In the discussion section of the results, the author mentioned that he used the experiment results just to test the theoretical results. Could you please explain this in the result section?

The discussion section was entirely rewritten.

The conclusion section looks like the abstract section and there is no real conclusion. The conclusion section should not contain reference number. Moreover, the discussion section should not contain reference number.

The conclusion section was entirely rewritten.

Reviewer 3 Report

The article focuses on enhancing industrial equipment vibration diagnostics using IoT-oriented solutions and Allan variance, based on the IEEE 1451.0-2007 standard.

Regrettably, the paper lacks scientific rigor and appears more like a laboratory work. Here are some suggestions to help the authors improve the paper:

 1.      The title should reflect that the paper also deals with IoT.

 2.      The Abstract needs better organization, including problem identification, proposed solutions, equipment, methods, results, and new perspectives.

 3.      The Introduction is overly extended, and the paper's aim was not properly assessed.

 4.      The paper contains basic knowledge presented in excessive detail, such as explanations of vibrational signals, vibration diagnosis, accelerometers, etc.

 5.      The paper's supposed innovative character relies on choosing energy-saving MEMS as sensors, but the scientific depth is still low.

 6.      It remains unclear whether the proposed method using Allan variance is genuinely new in vibration diagnosis. Instead of validating the results based on algorithm execution time and precision, the authors merely emphasize that the proposed equipment is energy-saving and already available on the market.

 7.      The paper should elaborate on how anomalous measurements are detected.

 8.      The conclusion that sensors were calibrated is not a significant scientific achievement. Additionally, the last conclusion lacks support from the numerical results presented in the paper.

The English is generally OK. The phrases are very clear and readable. 

Author Response

 1.      The title should reflect that the paper also deals with IoT.

The paper title was changed to “The Method of determining industrial equipment’s time and frequency vibration characteristics in the IoT system of condition-based maintenance.”

  2.      The Abstract needs better organization, including problem identification, proposed solutions, equipment, methods, results, and new perspectives.

The Abstract was entirely rewritten.

  3.      The Introduction is overly extended, and the paper’s aim was not properly assessed.

We took into account the comments and significantly changed the Introduction, better structured it, and somewhat shortened it, and at the end, we formulated the aim of the article.

  4.      The paper contains basic knowledge presented in excessive detail, such as explanations of vibrational signals, vibration diagnosis, accelerometers, etc.

We added the necessary explanations and cut down extra details, and the total volume of the Introduction was reduced by one and a half times.

  5.      The paper’s supposed innovative character relies on choosing energy-saving MEMS as sensors, but the scientific depth is still low.

I’m sorry, but the fourth reviewer (if there will be one) will write us a note that we are trying to squeeze two separate topics into one article.

  6.      It remains unclear whether the proposed method using Allan variance is genuinely new in vibration diagnosis. Instead of validating the results based on algorithm execution time and precision, the authors merely emphasize that the proposed equipment is energy-saving and already available on the market.

Added penultimate paragraph to the Introduction with links to the primary docs on the subject

  7.      The paper should elaborate on how anomalous measurements are detected.

The classical methods of removing anomalous measurements are well known, and we give the necessary references so as not to unnecessarily increase the length of the article.

  8.      The conclusion that sensors were calibrated is not a significant scientific achievement. Additionally, the last conclusion lacks support from the numerical results presented in the paper.

The conclusion was entirely rewritten.

Round 2

Reviewer 1 Report

Accept manuscript in present form

Author Response

We thank the reviewers for their generous comments on the manuscript and have edited the manuscript to address their concerns.

Reviewer 2 Report

The author did not do all my comments I mentioned for an example the introduction is still needs attention in which the author did not mention the name of the author in his introduction. The author did not add the nomenclature section. The author has been done some good results but he did not explain it in the right way. The same way with the conclusion section. I recommend that the author provide a rebuttal file beside the new version of the manuscript. 

Author Response

We thank the reviewers for their generous comments on the manuscript and have edited the manuscript to address their concerns. 

The author did not do all my comments I mentioned for an example the introduction is still needs attention in which the author did not mention the name of the author in his introduction.

Answer

The shortcomings of existing systems and the tasks that we have identified to overcome these shortcomings have been added to the "introduction" section. We did not mention names and backgrounds  of the authors in the introduction section of this article, since it is first article in these field of study

The author did not add the nomenclature section

Answer

Added section Abbreviations and Nomenclature

The author has been done some good results but he did not explain it in the right way. The same way with the conclusion section. I recommend that the author provide a rebuttal file beside the new version of the manuscript. 

The conclusions section has been rewritten to better describe the results obtained and the methods applied . A description of the next steps in the research and improvement of vibration diagnostic methods in CBM has also been added.

Reviewer 3 Report

The authors have made significant improvements to the paper. However, regrettably, the Conclusions section was nearly eliminated. I suggest reworking the conclusions section to highlight the main accomplishments, employed methods, and results.

The English used in this paper is OK.

Author Response

We thank the reviewers for their generous comments on the manuscript and have edited the manuscript to address their concerns. 

The authors have made significant improvements to the paper. However, regrettably, the Conclusions section was nearly eliminated. I suggest reworking the conclusions section to highlight the main accomplishments, employed methods, and results.

The conclusions section has been rewritten to better describe the results obtained and the methods applied . A description of the next steps in the research and improvement of vibration diagnostic methods in CBM has also been added.

Round 3

Reviewer 3 Report

The authors have answered all questions and taken all suggestions on board. I recommend the publication of this work in its present form.

The English is generally fine and only minor editing is necessary.